# DBGDGM: Dynamic Brain Graph Deep Generative Model

**Alexander Campbell**[*1,2]                                        AJRC4@CL.CAM.AC.UK

**Simeon Spasov**[*1,3]                                             SES88@CL.CAM.AC.UK

**Nicola Toschi** [4,5]

**Pietro Liò**[1]

[1] *Department of Computer Science and Technology, University of Cambridge, United Kingdom*

[2] *The Alan Turing Institute, United Kingdom*

[3] *German Center for Neurodegenerative Diseases (DZNE), Bonn, Germany*

[4] *University of Rome Tor Vergata, Italy*

[5] *A.A. Martinos Center for Biomedical Imaging, Harvard Medical School, United States*

**Editors:** Accepted for publication at MIDL 2023

## Abstract

Graphs are a natural representation of brain activity derived from functional magnetic imaging (fMRI) data. It is well known that clusters of anatomical brain regions, known as functional connectivity networks (FCNs), encode temporal relationships which can serve as useful biomarkers for understanding brain function and dysfunction. Previous works, however, ignore the temporal dynamics of the brain and focus on static graphs. In this paper, we propose a dynamic brain graph deep generative model (DBGDGM) which simultaneously clusters brain regions into temporally evolving communities and learns dynamic unsupervised node embeddings. Specifically, DBGDGM represents brain graph nodes as embeddings sampled from a distribution over communities that evolve over time. We parameterise this community distribution using neural networks that learn from subject and node embeddings as well as past community assignments. Experiments demonstrate DBGDGM outperforms baselines in graph generation, dynamic link prediction, and is comparable for graph classification. Finally, an analysis of the learnt community distributions reveals overlap with known FCNs reported in neuroscience literature. Code available at https://github.com/simeon-spasov/dynamic-brain-graph-deep-generative-model.

**Keywords:** Dynamic graph, generative model, functional magnetic resonance imaging

## 1. Introduction

Functional magnetic resonance imaging (fMRI) is a non-invasive imaging technique primarily used to measure blood-oxygen level dependent (BOLD) signal in the brain (Huettel et al., 2004). A natural representation of fMRI data is as a discrete-time graph, henceforth referred to as a dynamic brain graph (DBG), consisting of a set of fixed nodes corresponding to anatomically separated brain regions and a set of time-varying edges determined by a measure of dynamic functional connectivity (dFC) (Calhoun et al., 2014). DBGs have been widely used in graph-based network analysis for understanding brain function (Hirsch and Wohlschlaeger, 2022; Raz et al., 2016) and dysfunction (Alonso Martínez et al., 2020; Dautricourt et al., 2022; Yu et al., 2015).

---

Recently, there is growing interest in using deep learning-based methods for learning representations of graph-structured data (Goyal and Ferrara, 2018; Hamilton, 2020). A graph representation typically consists of a low-dimensional vector embedding of either the entire graph (Narayanan et al., 2017) or a part of it's structure such as nodes (Grover and Leskovec, 2016), edges (Gao et al., 2019), or sub-graphs (Adhikari et al., 2017). Although originally formulated for static graphs (i.e. not time-varying), several existing methods have been extended (Mahdavi et al., 2018; Goyal et al., 2020), and new ones proposed (Zhou et al., 2018; Sankar et al., 2020), for dynamic graphs. The embeddings are usually learnt in either a supervised or unsupervised fashion and typically used in tasks such as node classification (Pareja et al., 2020) and dynamic link prediction (Goyal et al., 2018).

To date, very few deep learning-based methods have been designed for, or existing methods applied to, representation learning of DBGs. Those that do, tend to use graph neural networks (GNNs) that are designed for learning node- and graph-level embeddings for use in graph classification (Kim et al., 2021; Dahan et al., 2021). Although node/graph-level embeddings are effective at representing local/global graph structure, they are less adept at representing topological structures in-between these two extremes such a clusters of nodes or communities (Wang et al., 2017). Recent methods that explicitly incorporate community embeddings alongside node embeddings have shown improved performance for static graph representation learning tasks (Sun et al., 2019; Cavallari et al., 2017). How to leverage the relatedness of graph, node, and community embeddings in a unified framework for DBG representation learning remains under-explored. We refer to Appendix A for a summary of related work.

**Contributions**  To address these shortcomings, we propose DBGDGM, a hierarchical deep generative model (DGM) designed for unsupervised representing learning on DBGs derived from multi-subject fMRI data. Specifically, DBGDGM represents nodes as embeddings sampled from a distribution over communities that evolve over time. The community distribution is parameterized using neural networks (NNs) that learn from graph and node embeddings as well as past community assignments. We evaluate DBGDGM on multiple real-world fMRI datasets and show that it outperforms state-of-the-art baselines for graph reconstruction, dynamic link prediction, and achieves comparable results for graph classification. Code on GitHub[1].

## 2. Related work

**Dynamic graph generative models**  Classic generative models for graph-structured data are designed for capturing a small set of specific properties (e.g. degree distribution, eigenvalues, modularity) of static graphs (Erdos et al., 1960; Barabási and Albert, 1999; Nowicki and Snijders, 2001). DGMs that exploit the learning capacity of NNs are able to learn more expressive graph distributions (Mehta et al., 2019; Kipf and Welling, 2016b; Sarkar et al., 2020). Recent DGMs for dynamic graphs are majority VAE-based (Kingma and Welling, 2013) and cannot learn community representations (Hajiramezanali et al., 2019; Gracious et al., 2021; Zhang et al., 2021). The few that do, are designed for static graphs (Sun et al., 2019; Khan et al., 2021; Cavallari et al., 2017).

---

1. https://github.com/simeon-spasov/dynamic-brain-graph-deep-generative-model

**Learning representations of dynamic brain graphs**  Unsupervised representation learning methods for DBGs tend to focus on clustering DBGs into a finite number of connectivity patterns that recur over time (Allen et al., 2014; Spencer and Goodfellow, 2022). Community detection is another commonly used method but mainly applied to static brain graphs (Pavlović et al., 2020; Esfahlani et al., 2021). Extensions to DBGs are typically not end-to-end trainable and do not scale to multi-subject datasets (Ting et al., 2020; Martinet et al., 2020b). Recent deep learning-based methods are predominately GNN-based (Kim et al., 2021; Dahan et al., 2021). Unlike DBGDGM, these methods are supervised and focus on learning deterministic node- and graph-level representations.

## 3. Problem formulation

We consider a dataset of multi-subject DBGs derived from fMRI data $\mathcal{D} \equiv \mathcal{G}^{(1:S, 1:T)} = \{\mathcal{G}^{(s,t)}\}_{s,t=1}^{S,T}$ that share a common set of nodes $\mathcal{V} = \{v_1, \ldots, v_V\}$ over $T \in \mathbb{N}$ timepoints for $S \in \mathbb{N}$ subjects. Each $\mathcal{G}^{(s,t)} \in \mathcal{G}^{(1:S, 1:T)}$ denotes a non-attributed, unweighted, and undirected brain graph snapshot for the $s$-th subject at the $t$-th timepoint. We define a brain graph snapshot as a tuple $\mathcal{G}^{(s,t)} = (\mathcal{V}, \mathcal{E}^{(s,t)})$ where $\mathcal{E}^{(s,t)} \subseteq \mathcal{V} \times \mathcal{V}$ denotes an edge set. The $i$-th edge for the $s$-th subject at the $t$-th timepoint $e_i^{(s,t)} \in \mathcal{E}^{(s,t)}$ is defined $e_i^{(s,t)} = (w_i^{(s,t)}, c_i^{(s,t)})$ where $w_i^{(s,t)}$ is a source node and $c_i^{(s,t)}$ is a target node. We assume each node corresponds to a brain region making the number of nodes $|\mathcal{V}| = V \in \mathbb{N}$ fixed over subjects and time. We also assume edges

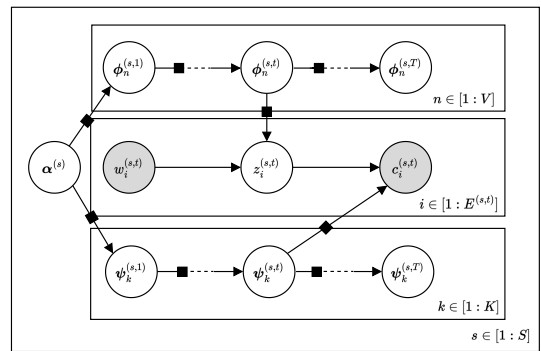

Figure 1: Plate diagram for DBGDGM. Latent and observed variables are denoted by white-and gray-shaded circles, respectively. Solid black squares denote non-linear mappings parameterized by NNs.

correspond to a measure of dFC allowing the number of edges $|\mathcal{E}^{(s,t)}| = E^{(s,t)} \in \mathbb{N}$ vary over subjects and time. We further assume there exists $K \in \mathbb{N}$ clusters of nodes, or communities, the membership of which dynamically changes over time for each subject. Let $z_i^{(s,t)} \in [1:K]$ denote the latent community assignment of the $i$-th edge for the $s$-th subject at the $t$-th timepoint. For each subject's DBG our aim is to learn, in an unsupervised fashion, graph $\boldsymbol{\alpha}^{(s)} \in \mathbb{R}^{H_\alpha}$, node $\boldsymbol{\phi}_{1:V}^{(s,t)} = [\boldsymbol{\phi}_n^{(s,t)}] \in \mathbb{R}^{V \times H_\phi}$, and community $\boldsymbol{\psi}_{1:K}^{(s,t)} = [\boldsymbol{\psi}_k^{(s,t)}] \in \mathbb{R}^{K \times H_\psi}$ representations of dimensions $H_\alpha, H_\phi, H_\psi \in \mathbb{N}$, respectively, for use in a variety of downstream tasks.

## 4. Method

DBGDGM defines a hierarchical deep generative model and inference network for the end-to-end learning of graph, node, and community embeddings from multi-subject DBG data. Specifically, DBGDGM treats the embeddings and edge community assignments as latent random variables collectively denoted $\Omega^{(s,t)} = \{\boldsymbol{\alpha}^{(s)}, \boldsymbol{\phi}_{1:V}^{(s,t)}, \boldsymbol{\psi}_{1:K}^{(s,t)}, \{z_i^{(s,t)}\}_{i=1}^{E^{(s,t)}}\}$, which

along with the observed DBGs, defines a probabilistic latent variable model with joint density $p_\theta(\mathcal{G}^{1:S,\,1:T}, \Omega^{1:S,\,1:T})$.

### 4.1. Generative model

**Graph embeddings**  We begin the generative process by sampling graph embeddings from a prior $\boldsymbol{\alpha}^{(s)} \sim p_{\theta_\alpha}(\boldsymbol{\alpha}^{(s)})$ implemented as a normal distribution following

$$p_{\theta_\alpha}(\boldsymbol{\alpha}^{(s)}) = \text{Normal}(\mathbf{0}_{H_\alpha}, \mathbf{I}_{H_\alpha}) \tag{1}$$

where $\mathbf{0}_{H_\alpha}$ is a matrix of zeros and $\mathbf{I}_{H_\alpha}$ is a identity matrix. Each embedding is a vector $\boldsymbol{\alpha}^{(s)} \in \mathbb{R}^{H_\alpha}$ representing subject-specific information that remains fixed over time.

**Node and community embeddings**  Next, let $\boldsymbol{\phi}_n^{(s,t)} \in \mathbb{R}^{H_\phi}$ and $\boldsymbol{\psi}_k^{(s,t)} \in \mathbb{R}^{H_\psi}$ denote the $n$-th node and the $k$-th community embedding, respectively. To incorporate temporal dynamics, we assume node and community embeddings are related through Markov chains with prior transition distributions $\boldsymbol{\phi}_n^{(s,t)} \sim p_{\theta_\phi}(\boldsymbol{\phi}_n^{(s,t)}|\boldsymbol{\phi}_n^{(s,t-1)})$ and $\boldsymbol{\psi}_k^{(s,t)} \sim p_{\theta_\psi}(\boldsymbol{\psi}_k^{(s,t)}|\boldsymbol{\psi}_k^{(s,t-1)})$. We specify each prior to be a normal distribution following

$$p_{\theta_\phi}(\boldsymbol{\phi}_n^{(s,t)}|\boldsymbol{\phi}_n^{(s,t-1)}) = \text{Normal}(\boldsymbol{\phi}_n^{(s,t-1)}, \sigma_\phi \mathbf{I}_{H_\phi}) \tag{2}$$

$$p_{\theta_\psi}(\boldsymbol{\psi}_k^{(s,t)}|\boldsymbol{\psi}_k^{(s,t-1)}) = \text{Normal}(\boldsymbol{\psi}_k^{(s,t-1)}, \sigma_\psi \mathbf{I}_{H_\psi}) \tag{3}$$

where the means are initialized via neural network transformations of the graph embeddings, i.e. $\boldsymbol{\phi}_n^{(s,0)} = \text{MLP}_{\theta_\phi}(\boldsymbol{\alpha}^{(s)})$, $\boldsymbol{\psi}_k^{(s,0)} = \text{MLP}_{\theta_\psi}(\boldsymbol{\alpha}^{(s)})$, where $\text{MLP}_{\theta_j} : \mathbb{R}^{H_\alpha} \to \mathbb{R}^{H_j}$ is a $L_j$-layered multilayer perceptron for $j \in \{\phi, \psi\}$. The standard deviations $\sigma_\phi, \sigma_\psi \in \mathbb{R}_{>0}$ are hyperparameters controlling how smoothly each embedding changes between consecutive snapshots.

**Edge generation**  We next describe the edge generative process of a graph snapshot $\mathcal{G}^{(s,t)} \in \mathcal{G}^{(1:S,\,1:T)}$. Similar to Sun et al. (2019), for each edge $e_i^{(s,t)} = (w_i^{(s,t)}, c_i^{(s,t)}) \in \mathcal{E}^{(s,t)}$ we first sample a latent community assignment $z_i^{(s,t)} \in [1:K]$ from a conditional prior $z_i^{(s,t)} \sim p_{\theta_z}(z_i^{(s,t)}|w_i^{(s,t)})$ implemented as a categorical distribution

$$p_{\theta_z}(z_i^{(s,t)}|w_i^{(s,t)}) = \text{Categorical}(\text{Softmax}(\tilde{\boldsymbol{\pi}}_i^{(s,t)})), \quad \tilde{\boldsymbol{\pi}}_i^{(s,t)} = \text{MLP}_{\theta_z}(\boldsymbol{\phi}_{w_i}^{(s,t)}) \tag{4}$$

where $\text{MLP}_{\theta_z} : \mathbb{R}^{H_\phi} \to \mathbb{R}^K$ is a $L_z$-layered MLP that parameterizes community probabilities using node embeddings indexed by $w_i^{(s,t)}$. In other words, each source node $w_i^{(s,t)}$ is represented as a mixture of communities. A linked target node $c_i^{(s,t)} \in [1:V]$ is then sampled from the conditional likelihood $c_i^{(s,t)} \sim p_{\theta_c}(c_i^{(s,t)}|z_i^{(s,t)})$ which is also implemented as a categorical distribution

$$p_{\theta_c}(c_i^{(s,t)}|z_i^{(s,t)}) = \text{Categorical}(\text{Softmax}(\hat{\boldsymbol{\pi}}_i^{(s,t)})), \quad \hat{\boldsymbol{\pi}}_i^{(s,t)} = \text{MLP}_{\theta_c}(\boldsymbol{\psi}_{z_i}^{(s,t)}) \tag{5}$$

where $\text{MLP}_{\theta_c} : \mathbb{R}^{H_\psi} \to \mathbb{R}^V$ is a $L_c$-layered MLP that parameterizes node probabilities using community embeddings indexed by $z_i^{(s,t)}$. That is, each community assignment $z_i^{(s,t)}$ is

represented as a mixture of nodes. By integrating out the latent community assignment variable

$$p(c_i^{(s,t)}|w_i^{(s,t)}) = \sum_{z_i^{(s,t)} \in [1:K]} p_{\theta_c}(c_i^{(s,t)}|z_i^{(s,t)})p_{\theta_z}(z_i^{(s,t)}|w_i^{(s,t)}) \tag{6}$$

we define the likelihood of node $c_i^{(s,t)}$ being a linked neighbor of node $w_i^{(s,t)}$, in a given graph snapshot.

**Factorized generative model**  Given this model specification, the joint probability of the observed data and the latent variables can be factorized following

$$p_\theta(\mathcal{G}^{1:S\,1:T}, \Omega^{1:S,1:T}) = \prod_{s=1}^{S} \Bigg( p_{\theta_\alpha}(\boldsymbol{\alpha}^{(s)}) \prod_{t=1}^{T} \Bigg( \prod_{n=1}^{V} p_{\theta_\phi}(\boldsymbol{\phi}_n^{(s,t)}|\boldsymbol{\phi}_n^{(s,t-1)})$$
$$\prod_{k=1}^{K} p_{\theta_\psi}(\boldsymbol{\psi}_k^{(s,t)}|\boldsymbol{\psi}_k^{(s,t-1)}) \tag{7}$$
$$\prod_{i=1}^{E^{(s,t)}} p_{\theta_z}(z_i^{(s,t)}|\boldsymbol{\phi}_{w_i}^{(s,t)})p_{\theta_c}(c_i^{(s,t)}|\boldsymbol{\psi}_{z_i}^{(s,t)}) \Bigg) \Bigg)$$

where $\theta = \{\theta_\phi, \theta_\psi, \theta_z, \theta_c\}$ is the set of generative model parameters i.e. NN weights. The generative model of DBGDGM summarized in Appendix B.

## 4.2. Inference network

Inferring the posterior $p_\theta(\Omega^{(1:S,1:T)}|\mathcal{G}^{(1:S,1:T)})$ is intractable so we resort to variational inference (Jordan et al., 1999) to approximate the true posterior with a variational distribution $q_\lambda(\Omega^{(1:S,1:T)})$ with parameters $\lambda$. For training, we maximize a lower bound on the log marginal likelihood of the DBGs, referred to as the ELBO (**e**vidence **l**ower **bo**und):

$$\mathcal{L}_{\text{ELBO}}(\theta, \lambda) = \mathbb{E}_{q_\lambda}\Bigg[ \log \frac{p_\theta(\mathcal{G}^{1:S,\,1:T}, \Omega^{1:S,\,1:T})}{q_\lambda(\Omega^{(1:S,1:T)})} \Bigg] \le \log p_\theta(\mathcal{G}^{(1:S,\,1:T)}) \tag{8}$$

where $\mathbb{E}_{q_\lambda}[\cdot]$ denotes the expectation with respect to the variational distribution $q_\lambda(\Omega^{(1:S,\,1:T)})$. By maximizing the ELBO with respect to the generative and variational parameters $\theta$ and $\lambda$ we train our generative model and perform Bayesian inference, respectively.

**Structured variational distribution**  To ensure a good approximation to true the posterior, we retain the Markov properties of the node and community embeddings. This results in a structured variational distribution (Hoffman and Blei, 2015; Saul and Jordan, 1995) which factorizes following

$$q_\lambda(\Omega^{(1:S,1:T)}) = \prod_{s=1}^{S} \Bigg( q_{\lambda_\alpha}(\boldsymbol{\alpha}^{(s)}) \prod_{t=1}^{T} \Bigg( \prod_{n=1}^{V} q_{\lambda_\phi}(\boldsymbol{\phi}_n^{(s,t)}|\boldsymbol{\phi}_n^{(s,t-1)})$$
$$\prod_{k=1}^{K} q_{\lambda_\psi}(\boldsymbol{\psi}_k^{(s,t)}|\boldsymbol{\psi}_k^{(s,t-1)}) \prod_{i=1}^{E^{(s,t)}} q_{\lambda_z}(z_i^{(s,t)}|\boldsymbol{\phi}_{w_i}^{(s,t)}, \boldsymbol{\phi}_{c_i}^{(s,t)}) \Bigg) \Bigg). \tag{9}$$

Moreover, each distribution is specified to mimic the structure of the generative model such that

$$q_{\lambda_\alpha}(\boldsymbol{\alpha}^{(s)}) = \text{Normal}(\boldsymbol{\mu}^{(s)}, \boldsymbol{\sigma}^{(s)}) \tag{10}$$

$$q_{\lambda_\phi}(\boldsymbol{\phi}_n^{(s,t)}|\boldsymbol{\phi}_n^{(s,t-1)}) = \text{Normal}(\tilde{\boldsymbol{\mu}}_n^{(s,t)}, \tilde{\boldsymbol{\sigma}}_n^{(s,t)}), \quad \{\tilde{\boldsymbol{\mu}}_n^{(s,t)}, \tilde{\boldsymbol{\sigma}}_n^{(s,t)}\} = \text{GRU}_{\lambda_\phi}(\boldsymbol{\phi}_n^{(s,t-1)}) \tag{11}$$

$$q_{\lambda_\psi}(\boldsymbol{\psi}_k^{(s,t)}|\boldsymbol{\psi}_k^{(s,t-1)}) = \text{Normal}(\hat{\boldsymbol{\mu}}_k^{(s,t)}, \hat{\boldsymbol{\sigma}}_k^{(s,t)}), \quad \{\hat{\boldsymbol{\mu}}_k^{(s,t)}, \hat{\boldsymbol{\sigma}}_k^{(s,t)}\} = \text{GRU}_{\lambda_\psi}(\boldsymbol{\psi}_k^{(s,t-1)}) \tag{12}$$

$$q_{\lambda_z}(z_i^{(s,t)}|\boldsymbol{\phi}_{w_i}^{(s,t)}, \boldsymbol{\phi}_{c_i}^{(s,t)}) = \text{Categorical}(\text{Softmax}(\boldsymbol{\pi}_i^{(s,t)})), \quad \boldsymbol{\pi}_i^{(s,t)} = \text{MLP}_{\lambda_z}(\boldsymbol{\phi}_{w_i}^{(s,t)} \odot \boldsymbol{\phi}_{c_i}^{(s,t)}) \tag{13}$$

where $\text{GRU}_{\lambda_j} : \mathbb{R}^{H_j} \to \mathbb{R}^{H_j}$ is a $L_j$-layered GRU for each $j \in \{\phi, \psi\}$ and $\text{MLP}_{\lambda_z} : \mathbb{R}^{H_\phi} \to \mathbb{R}^K$ is $L_z$-layered MLP. Furthermore, we use MLPs to initialize the GRUs with the graph embeddings such that $\boldsymbol{\phi}_n^{(s,0)} = \text{MLP}_{\lambda_\phi}(\boldsymbol{\alpha}^{(s)})$ and $\boldsymbol{\psi}_k^{(s,0)} = \text{MLP}_{\lambda_\psi}(\boldsymbol{\alpha}^{(s)})$ where $\text{MLP}_{\lambda_j} : \mathbb{R}^{N_\alpha} \to \mathbb{R}^{N_j}$. This allows for subject-specific variation to be incorporated in the temporal dynamics of the node and community embeddings. Another difference with the generative model is now the variational distribution of the community assignment $q_{\lambda_z}(\cdot)$ includes information from neighboring nodes via $c_i^{(s,t)}$. Finally, we use the same NNs from the generative model to parameterize the variational distributions for the node and community embeddings as well as the community assignment. This not only spares additional trainable parameters for the variational distribution but also further links the variational parameters of $q_\lambda(\cdot)$ to generative parameters of $p_\theta(\cdot)$ resulting in more robust learning (Farnoosh and Ostadabbas, 2021). The set of parameters for the inference network is therefore $\lambda = \{\lambda_\alpha = \{\boldsymbol{\mu}^{(s)}, \boldsymbol{\sigma}^{(s)}\}_{s=1}^S, \lambda_\phi = \theta_\phi, \lambda_\psi = \theta_\psi, \lambda_z = \theta_z\}$.

**Training objective** Substituting the variational distribution from (9) and the joint distribution from (7) into the ELBO (8) gives the full training objective which can be optimized using stochastic gradient descent. We estimate all gradients using the reparameterization trick (Kingma and Welling, 2013) and the Gumbel-softmax trick (Jang et al., 2016; Maddison et al., 2016). We refer to Appendix B for further details on the ELBO derivation and learning the parameters.

## 5. Experiments

We evaluate DBGDGM against baseline models on the tasks of graph reconstruction, dynamic link prediction, and graph classification. Each task is designed to evaluate the usefulness of the learnt embeddings.

**Datasets** We construct two multi-subject DBG datasets using publicly available fMRI data from the Human Connectome Project (HCP) (Van Essen et al., 2013) and UK Biobank (UKB) (Sudlow et al., 2015). We randomly sample $S = 300$ subjects ensuring an even biological sex split. To create DBGs, we parcellate each scan into $V = 360$ region-wise BOLD signals using the Glasser atlas (Glasser et al., 2016), apply sliding-window Pearson correlation (Calhoun et al., 2014) with a non-overlapping window of size and stride of 30, and threshold the top 5% values of the lower triangle of each correlation matrix as connected

| Model | HCP | | UKB | |
|---|---|---|---|---|
| | **NLL** ($\downarrow$) | **MSE** ($\downarrow$) | **NLL** ($\downarrow$) | **MSE** ($\downarrow$) |
| CMN | $5.999 \pm 0.029$ * | $0.050 \pm 0.005$ * | $5.861 \pm 0.017$ * | $0.050 \pm 0.003$ * |
| VGAE | $5.857 \pm 0.017$ * | $0.051 \pm 0.002$ * | $5.851 \pm 0.027$ * | $0.061 \pm 0.002$ * |
| OSBM | $5.808 \pm 0.026$ * | $0.051 \pm 0.003$ * | $5.726 \pm 0.039$ * | $0.052 \pm 0.003$ * |
| VGRAPH | $\underline{5.569 \pm 0.046}$ * | $0.022 \pm 0.004$ * | $5.716 \pm 0.037$ * | $0.020 \pm 0.003$ * |
| VGRNN | $5.674 \pm 0.034$ * | $\underline{0.011 \pm 0.003}$ * | $\underline{5.649 \pm 0.035}$ * | $\underline{0.014 \pm 0.002}$ * |
| ELSM | $5.924 \pm 0.040$ * | $0.081 \pm 0.002$ * | $5.809 \pm 0.024$ * | $0.115 \pm 0.003$ * |
| DBGDGM | $\mathbf{4.587 \pm 0.045}$ | $\mathbf{0.001 \pm 0.002}$ | $\mathbf{4.586 \pm 0.084}$ | $\mathbf{0.004 \pm 0.003}$ |
| | **AUROC** ($\uparrow$) | **AP** ($\uparrow$) | **AUROC** ($\uparrow$) | **AP** ($\uparrow$) |
| CMN | $0.665 \pm 0.007$ * | $0.654 \pm 0.006$ * | $0.678 \pm 0.004$ * | $0.668 \pm 0.005$ * |
| VGAE | $0.661 \pm 0.010$ * | $0.674 \pm 0.008$ * | $0.688 \pm 0.010$ * | $0.607 \pm 0.009$ * |
| OSBM | $0.655 \pm 0.027$ * | $0.675 \pm 0.024$ * | $0.678 \pm 0.032$ * | $0.682 \pm 0.033$ * |
| VGRAPH | $\underline{0.689 \pm 0.004}$ * | $0.682 \pm 0.002$ * | $0.664 \pm 0.002$ * | $0.621 \pm 0.001$ * |
| VGRNN | $\underline{0.689 \pm 0.007}$ * | $\underline{0.698 \pm 0.006}$ * | $\underline{0.698 \pm 0.009}$ * | $\underline{0.696 \pm 0.007}$ * |
| ELSM | $0.669 \pm 0.004$ * | $0.662 \pm 0.002$ * | $0.661 \pm 0.001$ * | $0.662 \pm 0.002$ * |
| DBGDGM | $\mathbf{0.768 \pm 0.026}$ | $\mathbf{0.732 \pm 0.032}$ | $\mathbf{0.786 \pm 0.040}$ | $\mathbf{0.762 \pm 0.038}$ |

Table 1: Graph reconstruction (top) and dynamic link prediction (bottom) results (mean $\pm$ standard deviation over 5 runs). First and second-best results shown in **bold** and underlined. Statistically significant difference from DBGDGM marked *.

following Kim et al. (2021). The described procedure gives $T = 16$ graph snapshots for each subject. Biological sex is taken as graph-level labels. We refer to Appendix D for further details on each dataset.

**Baselines**   We compare DBGDGM against a range of different unsupervised probabilistic baseline models. For static baselines, we include variational graph autoencoder (VGAE) (Kipf and Welling, 2016b), a deep generative version of the overlapping stochastic block model (OSBM) (Mehta et al., 2019), and vGraph (VGRAPH) (Sun et al., 2019). For dynamic baselines we include variational graph recurrent neural network (VGRNN) (Hajiramezanali et al., 2019) and evolving latent space model (ELSM) (Gupta et al., 2019). For the graph reconstruction and link prediction tasks, we also include a heuristic baseline based on common neighbors between nodes at previous snapshots (CMN). Finally, for graph classification we include a support vector machine which takes as inputs static FC matrices (FCM) (Abraham et al., 2017). Further details about baseline models can be found in Appendix E.

**Implementation**   For training the unsupervised models, we split both datasets into 80/10/10% training/validation/test data along the time dimension. We train all models using the Adam optimizer (Kingma and Ba, 2014) with decoupled weight decay (Loshchilov and Hutter, 2017). All baseline hyperparameters are set following their original implementations. For DBGDGM, we choose the number of communities $K$ based on validation NLL. Finally, we train all models 5 times using different random seeds for 1,000 iterations and save the model with lowest validation NLL. Implementation details can be found in Appendix F.

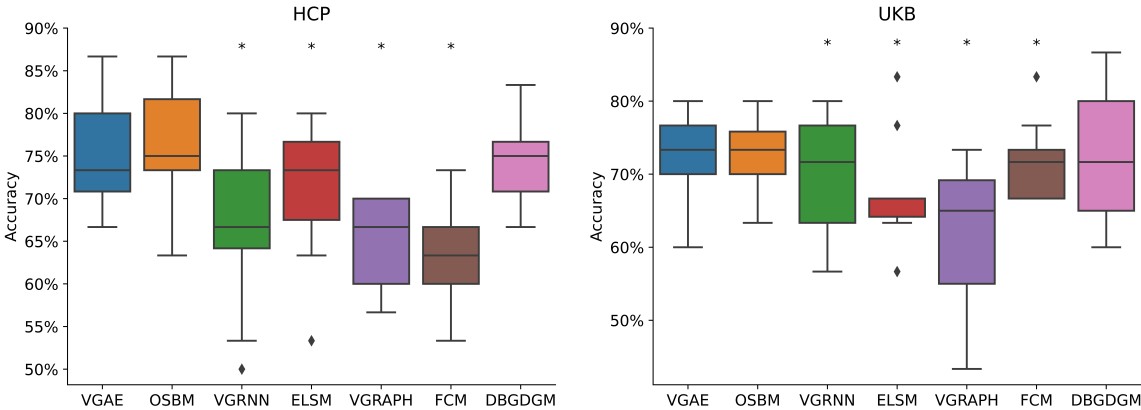

Figure 2: Graph classification results (5 runs). Statistical significance from DBGDGM marked *.

**Evaluation metrics** We evaluate the learnt embeddings on a variety of downstream tasks. For graph reconstruction and dynamic link prediction, we assess performance on the test edges produced by the 80/10/10% temporal split. We calculate the probability of the test edges using negative log-likelihood (NLL) and also compare the mean-squared error (MSE) between actual and reconstructed node degree over all test snapshots. For dynamic link prediction, we sample an equal number of positive and negative edges in the test dataset and measure performance using area under the receiver operator curve (AUROC) and average precision (AP).Finally, for graph classification, we predict the biological sex for each subjects' DBG and evaluate on accuracy. To predict graph labels, we average node embeddings per subject for the baselines and the community embeddings for DBGDGM. The averaged embeddings are split subject-wise into 80/20% train/test datasets and an SVM is trained using 10-fold cross validation on the train split. This is repeated 5 times using a different random seed. For comparing models, we use the almost stochastic order (ASO) test (Dror et al., 2019) with significance level 0.05 and correct for multiple comparisons (Bonferroni, 1936). A description of each task is included in Appendix G.

## 6. Results

**Dynamic graph reconstruction and link prediction.** We summarize the average test results of all models over 5 runs using optimally tuned hyperparameters.From Table 1, it is clear that DBGDGM outperforms baselines on both tasks. For graph reconstruction, DBGDGM shows an 18% and 30% relative improvement in NLL on HCP and UKB, respectively, compared to the second-best baselines. For dynamic link prediction, the relative improvement is $> 11\%$ in AUCROC and $> 5\%$ in AP compared to second-best baselines depending on dataset. We attribute these statistically significant gains to DBGDGM's ability to learn dynamic brain connectivity more effectively.

**Graph classification** For graph classification, DBGDGM achieves $\sim 75\%$ accuracy for HCP and $\sim 73\%$ for UKB (see Fig. 2). We outperform 4 baselines and show indiscernible

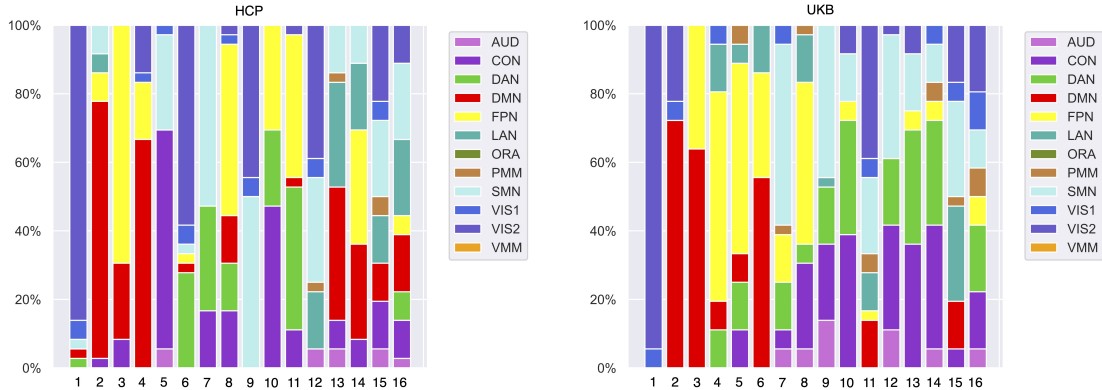

Figure 3: Overlap between communities learned by DBGDGM and FCNs from Ji et al. (2019). Some communities fully correspond to known FCNs, others are a mix, and offer a way to study FCN co-activation.

performance to VGAE and OSBM. To show the interpretative power of DBGDGM, we re-run the graph classification experiment for HCP with the embeddings of each community separately. We find a community which comprises brain regions in the Cingulo-opercular (CON) and the Somatomotor (SMN) networks, which achieves 68% accuracy. This finding is in agreement with studies that show SMN is predictive of gender (Zhang et al., 2018). With the exception of VGRAPH, which DBGDGM outperforms, such an interpretability analysis cannot be done in a computationally feasible way by any of the other baselines.

**Interpretability analysis** We use the learnt distributions over the nodes to calculate overlap between each community and known functional connectivity networks (FCNs) from Ji et al. (2019) (see Appendix H). Figure 3 shows that DBGDGM finds communities that significantly overlap with existing FCNs. In particular, nodes in community 1 almost fully corresponds to the visual network (VIS1 + VIS2), which is in keeping with the nature of the experiment (the resting state data was acquired with eyes open and cross-hair fixation). Remarkably, the second and third most homogeneous communities correspond to a large degree to the DMN, which is well known to dominate resting state activity as a whole (Yeshurun et al., 2021). The inspection of additional communities and respective predictive power, along with their evolution in time has the potential to unveil the relationships between dynamic brain connectivity changes and, e.g. psychiatric or neurological disorders.

## 7. Conclusion

We propose DBGDGM, a hierarchical DGM designed for unsupervised representation learning of DBGs. Specifically, DBGDGM jointly learns graph-, community-, and node-level embeddings that outperform baselines on classification, interpretability, and dynamic link prediction with statistical significance. Moreover, an analysis of the learnt dynamic community-node distributions shows significant overlap with existing FCNs from neuroscience literature further validating our method.

## Acknowledgments

This work is supported by The Alan Turing Institute under the EPSRC grant EP/N510129/1. Data were provided [in part] by the Human Connectome Project, WU-Minn Consortium (Principal Investigators: David Van Essen and Kamil Ugurbil; 1U54MH091657) funded by the 16 NIH Institutes and Centers that support the NIH Blueprint for Neuroscience Research; and by the McDonnell Center for Systems Neuroscience at Washington University. The UK Biobank data (application 20904) were curated and analyzed using a computational facility funded by an MRC research infrastructure award (MR/M009041/1) and supported by the NIHR Cambridge Biomedical Research Centre and a Marmaduke Shield Award to Dr. Richard A.I. Bethlehem and Varun Warrier. The views expressed are those of the authors and not necessarily those of the NHS, the NIHR or the Department of Health and Social Care.

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

## Appendix A. Related work

**Dynamic graph generative models** Classic generative models for graph-structured data are designed for capturing a small set of specific properties (e.g. degree distribution, eigenvalues, modularity) of static graphs (Erdos et al., 1960; Barabási and Albert, 1999; Nowicki and Snijders, 2001). DGMs that exploit the learning capacity of NNs are able to learn more expressive graph distributions (Mehta et al., 2019; Kipf and Welling, 2016b; Sarkar et al., 2020). Recent DGMs for dynamic graphs are majority VAE-based (Kingma and Welling, 2013) and cannot learn community representations (Hajiramezanali et al., 2019; Gracious et al., 2021; Zhang et al., 2021). The few that do, are designed for static graphs (Sun et al., 2019; Khan et al., 2021; Cavallari et al., 2017).

**Learning representations of dynamic brain graphs** BOLD signals derived from fMRI, whether at the voxel or brain region level, represent non stationary timeseries (Guan et al., 2020). As such, how the signals relate to each other spatially changes over time. Within the context of dynamic functional connectivity, it is essential to capture these time varying spatial relationships. Unsupervised representation learning methods for DBGs tend to focus on clustering DBGs into a finite number of connectivity patterns that recur over time (Allen et al., 2014; Spencer and Goodfellow, 2022). Community detection is another commonly used method but mainly applied to static brain graphs (Pavlović et al., 2020; Esfahlani et al., 2021). Extensions to DBGs are typically not end-to-end trainable and do not scale to multi-subject datasets (Ting et al., 2020; Martinet et al., 2020b). Recent deep learning-based methods are predominately GNN-based (Kim et al., 2021; Dahan et al., 2021). Unlike DBGDGM, these methods are supervised and focus on learning deterministic node- and graph-level representations.

## Appendix B. Method

### B.1. Generative model

Algorithm 1 summarizes the generative model for DBGDGM.

### B.2. Training objective and learning the parameters

Substituting the variational distribution from (9) and the joint distribution from (7) into the ELBO (8) gives the full training objective defined as

$$
\begin{aligned}
\mathcal{L}_{\mathrm{ELBO}}(\theta, \lambda) = \sum_{s=1}^{S} \sum_{t=1}^{T} \sum_{i=1}^{E^{(s,t)}} & \Bigg( \mathbb{E}_{q_{\lambda_z} q_{\lambda_\psi}} \Big[ \log p_{\theta_c}(c_i^{(s,t)} | w_i^{(s,t)}, \psi_{z_i}^{(s,t)}) \Big] \\
& - \mathbb{E}_{q_{\lambda_\phi}} \Big[ \mathrm{D_{KL}}[q_{\lambda_z}(z_i^{(s,t)} | \phi_{w_i}^{(s,t)}, \phi_{c_i}^{(s,t)}) || p_{\theta_z}(z_i^{(s,t)} | \phi_{w_i}^{(s,t)})] \Big] \Bigg) \\
& - \sum_{s=1}^{S} \Bigg( \mathrm{D_{KL}}[q_{\lambda_\alpha}(\boldsymbol{\alpha}^{(s)}) || p_{\theta_\alpha}(\boldsymbol{\alpha}^{(s)})] \sum_{t=1}^{T} \Bigg( \\
& - \sum_{n=1}^{V} \mathbb{E}_{q_{\lambda_\phi}} \Big[ \mathrm{D_{KL}}[q_{\lambda_\phi}(\phi_n^{(s,t)} | \phi_n^{(s,t-1)}) || p_{\theta_\phi}(\phi_n^{(s,t)} | \phi_n^{(s,t-1)})] \Big] \\
& - \sum_{k=1}^{K} \mathbb{E}_{q_{\lambda_\psi}} \Big[ \mathrm{D_{KL}}[q_{\lambda_\psi}(\psi_k^{(s,t)} | \psi_k^{(s,t-1)}) || p_{\theta_\psi}(\psi_k^{(s,t)} | \psi_k^{(s,t-1)})] \Big] \Bigg) \Bigg)
\end{aligned}
\tag{14}
$$

where $\mathrm{D_{KL}}[\cdot || \cdot]$ denotes the Kullback-Leibler (KL) divergence. By maximizing (14), the parameters $(\theta, \lambda)$ of the generative model and inference network can be jointly learnt.

**Learning the parameters** In order to use efficient stochastic gradient-based optimization techniques (Robbins and Monro, 1951) for learning $(\theta, \lambda)$, the gradient of the ELBO has to be estimated. The main challenge of this is obtaining gradients of the variables under expectation, i.e. $\mathbb{E}_{q_*}[\cdot]$, since they are sampled. To allow gradients to flow through these sampling steps, we use the reparameterization trick (Kingma and Welling, 2013; Rezende

---

**Algorithm 1:** DBGDGM generative model

---

**Input:** Common node set $\mathcal{V}$, source nodes from all edges
$\quad\quad \{w_i^{(s,t)} : i = 1, \ldots, E^{(s,t)}\}_{s,t=1}^{S,T}$

**Hyperparameters:** Number of communities $K$; embedding dimensions $H_\alpha$, $H_\phi$, $H_\psi$;
$\quad\quad\quad\quad\quad\quad$ number of layers in NNs $L_\phi$, $L_\psi$, $L_z$; temporal smoothness $\sigma_\psi$, $\sigma_\phi$

---

**Initialize:** $\mathcal{D} \leftarrow \varnothing$

**for** $s \leftarrow 1$ **to** $S$ **do**

$\quad$ /* Initialize node and community embeddings */

$\quad$ /* Sample graph embeddings from prior */

$\quad \boldsymbol{\alpha}^{(s)} \sim p(\boldsymbol{\alpha}^{(s)}) = \text{Normal}(\mathbf{0}_{H_\alpha}, \mathbf{I}_{H_\alpha})$

$\quad$ **for** $t \leftarrow 1$ **to** $T$ **do**

$\quad\quad$ **for** $k \leftarrow 1$ **to** $K$ **do**

$\quad\quad\quad$ /* Sample community embeddings from prior */

$\quad\quad\quad \boldsymbol{\psi}_k^{(s,0)} = \text{MLP}_{\theta_\psi}(\boldsymbol{\alpha}^{(s)})$

$\quad\quad\quad \boldsymbol{\psi}_k^{(s,t)} \sim p(\boldsymbol{\psi}_k^{(s,t)}|\boldsymbol{\psi}_k^{(s,t-1)}) = \text{Normal}(\boldsymbol{\psi}_k^{(s,t-1)}, \sigma_\psi \mathbf{I}_{H_\psi})$

$\quad\quad$ **end**

$\quad\quad$ **for** $n \leftarrow 1$ **to** $V$ **do**

$\quad\quad\quad$ /* Sample node embeddings from prior */

$\quad\quad\quad \boldsymbol{\phi}_n^{(s,0)} = \text{MLP}_{\theta_\phi}(\boldsymbol{\alpha}^{(s)})$

$\quad\quad\quad \boldsymbol{\phi}_n^{(s,t)} \sim p(\boldsymbol{\phi}_n^{(s,t)}|\boldsymbol{\phi}_n^{(s,t-1)}) = \text{Normal}(\boldsymbol{\phi}_n^{(s,t-1)}, \sigma_\phi \mathbf{I}_{H_\phi})$

$\quad\quad$ **end**

$\quad\quad \mathcal{E}^{(s,t)} \leftarrow \varnothing$

$\quad\quad$ **for** $i \leftarrow 1$ **to** $E^{(s,t)}$ **do**

$\quad\quad\quad$ /* Sample community assignment from prior */

$\quad\quad\quad \tilde{\boldsymbol{\pi}}_i^{(s,t)} = \text{MLP}_{\theta_z}(\boldsymbol{\phi}_{w_i}^{(s,t)})$

$\quad\quad\quad z_i^{(s,t)} \sim p(z_i^{(s,t)}|w_i^{(s,t)}) = \text{Categorical}(\text{Softmax}(\tilde{\boldsymbol{\pi}}_i^{(s,t)}))$

$\quad\quad\quad$ /* Sample linked neighbor from conditional likelihood */

$\quad\quad\quad \hat{\boldsymbol{\pi}}_i^{(s,t)} = \text{MLP}_{\theta_c}(\boldsymbol{\psi}_{z_i}^{(s,t)})$

$\quad\quad\quad c_i^{(s,t)} \sim p_{\theta_c}(c_i^{(s,t)}|z_i^{(s,t)}) = \text{Categorical}(\text{Softmax}(\hat{\boldsymbol{\pi}}_i^{(s,t)}))$

$\quad\quad\quad \mathcal{E}^{(s,t)} \leftarrow \mathcal{E}^{(s,t)} \cup \{(w_i^{(s,t)}, c_i^{(s,t)})\}$

$\quad\quad$ **end**

$\quad\quad \mathcal{G}^{(s,t)} \leftarrow (\mathcal{V}, \mathcal{E}^{(s,t)})$

$\quad\quad \mathcal{D} \leftarrow \mathcal{D} \cup \{\mathcal{G}^{(s,t)}\}$

$\quad$ **end**

**end**

---

et al., 2014) for the normal distributions and the Gumbel-softmax trick (Jang et al., 2016; Maddison et al., 2016) for the categorical distributions. All gradients are now easily computed via back-propagation (Rumelhart et al., 1986) making DBGDGM end-to-end trainable. In addition, we analytically calculate the KL terms for both normal and categorical distributions, which leads to lower variance gradient estimates and faster training as com-

---

**Algorithm 2:** DBGDGM training

---

**Input:** Common node set $\mathcal{V}$, source nodes from all edges
$\quad\quad \{w_i^{(s,t)} : i = 1, \ldots, E^{(s,t)}\}_{s,t=1}^{S,T}$
**Hyperparameters:** Number of communities $K$; embedding dimensions $H_\alpha$, $H_\phi$, $H_\psi$;
$\quad\quad\quad\quad\quad\quad$ number of layers in NNs $L_\phi$, $L_\psi$ $L_z$

**Initialize:** $\{\boldsymbol{\mu}^{(s)}, \log\boldsymbol{\sigma}^{(s)}\}_{s=1}^{S} \leftarrow \text{Normal}(\mathbf{0}, \mathbf{1})$
**for** $s \leftarrow 1$ **to** $S$ **do**
$\quad$ /* Sample graph embeddings from posterior */
$\quad$ $\boldsymbol{\alpha}^{(s)} \sim q_{\lambda_\alpha}(\boldsymbol{\alpha}^{(s)}) = \text{Normal}(\boldsymbol{\mu}^{(s)}, \boldsymbol{\sigma}^{(s)})$
$\quad$ **for** $t \leftarrow 1$ **to** $T$ **do**
$\quad\quad$ **for** $k \leftarrow 1$ **to** $K$ **do**
$\quad\quad\quad$ /* Sample community embeddings from posterior */
$\quad\quad\quad$ $\{\tilde{\boldsymbol{\mu}}_k^{(s,t)}, \tilde{\boldsymbol{\sigma}}_k^{(s,t)}\} = \text{GRU}_{\lambda_\psi}(\boldsymbol{\psi}_k^{(s,t-1)}), \quad \boldsymbol{\psi}_k^{(s,0)} = \text{MLP}_{\lambda_\psi}(\boldsymbol{\alpha}^{(s)})$
$\quad\quad\quad$ $\boldsymbol{\psi}_k^{(s,t)} \sim q_{\lambda_\psi}(\boldsymbol{\psi}_k^{(s,t)}|\boldsymbol{\psi}_k^{(s,t-1)}) = \text{Normal}(\tilde{\boldsymbol{\mu}}_k^{(s,t)}, \tilde{\boldsymbol{\sigma}}_k^{(st)})$
$\quad\quad$ **end**
$\quad\quad$ **for** $n \leftarrow 1$ **to** $V$ **do**
$\quad\quad\quad$ /* Sample node embeddings from posterior */
$\quad\quad\quad$ $\{\hat{\boldsymbol{\mu}}_n^{(s,t)}, \hat{\boldsymbol{\sigma}}_n^{(s,t)}\} = \text{GRU}_{\lambda_\phi}(\boldsymbol{\phi}_n^{(s,t-1)}), \quad \boldsymbol{\phi}_n^{(s,0)} = \text{MLP}_{\lambda_\phi}(\boldsymbol{\alpha}^{(s)})$
$\quad\quad\quad$ $\boldsymbol{\phi}_n^{(s,t)} \sim q_{\lambda_\phi}(\boldsymbol{\phi}_n^{(s,t)}|\boldsymbol{\phi}_n^{(s,t-1)}) = \text{Normal}(\hat{\boldsymbol{\mu}}_n^{(s,t)}, \hat{\boldsymbol{\sigma}}_n^{(st)})$
$\quad\quad$ **end**
$\quad\quad$ **for** $i \leftarrow 1$ **to** $E^{(s,t)}$ **do**
$\quad\quad\quad$ /* Sample community assignment from posterior */
$\quad\quad\quad$ $\boldsymbol{\pi}_i^{(s,t)} = \text{MLP}_{\lambda_z}(\boldsymbol{\phi}_{w_i}^{(s,t)} \odot \boldsymbol{\phi}_{c_i}^{(s,t)})$
$\quad\quad\quad$ $z_i^{(s,t)} \sim p_{\lambda_z}(z_i^{(s,t)}|w_i^{(s,t)}, c_i^{(s,t)}) = \text{Categorical}(\boldsymbol{\pi}_i^{(s,t)})$
$\quad\quad$ **end**
$\quad$ **end**
**end**

---

pared to noisy Monte Carlo estimates. Algorithm 2 summarizes all steps of the training procedure.

## B.3. Table of notation

Table 2 summarizes the notation used in this paper.

| Notation | Description |
|---|---|
| $S$ | Number of subjects. |
| $T$ | Number of timepoints. |
| $\mathcal{G}^{(1:S,1:T)} = \{\mathcal{G}^{(s,t)}\}_{s,t=1}^{S,T}$ | Multi-subject dynamic brain graph dataset derived from fMRI. |
| $\mathcal{G}^{(s,t)} = (\mathcal{V}, \mathcal{E}^{(s,t)})$ | Dynamic brain graph snapshot of the $s$-th subject at the $t$-th timepoint. |
| $\mathcal{V} = \{v_1, \ldots, v_V\}$ | Set of common nodes. |
| $V$ | Number of nodes. |
| $\mathcal{E}^{(s,t)} \subseteq \mathcal{V} \times \mathcal{V}$ | Edge set. |
| $(w_i^{(s,t)}, c_i^{(s,t)})$ | Source node and target node of the $i$-th edge. |
| $E^{(s,t)}$ | Number of edges. |
| $K$ | Number of communities. |
| $\boldsymbol{\alpha}^{(s)} \in \mathbb{R}^{N_\alpha}$ | Subject embedding of dimensionality $H_\alpha$. |
| $\boldsymbol{\phi}_n^{(s,t)} \in \mathbb{R}^{H_\phi}$ | Node embedding of dimensionality $H_\phi$. |
| $\boldsymbol{\psi}_k^{(s,t)} \in \mathbb{R}^{H_\psi}$ | Community embedding of dimensionality $H_\psi$. |
| $z_i^{(s,t)} \in [1:K]$ | Edge community assignment. |
| $\Omega^{(s,t)}$ | Set of latent variables, i.e., $\boldsymbol{\alpha}^{(s)}, \boldsymbol{\phi}^{(s,t)}, \boldsymbol{\psi}^{(s,t)}, \{z_i^{(s,t)}\}_{i=1}^{E^{(s,t)}} \in \Omega^{(s,t)}$. |
| $p_\theta(\mathcal{G}^{(1:S,1:T)}, \Omega^{(1:S,1:T)})$ | Joint distribution of observed dynamic brain graphs and unobserved latent variables, i.e., generative model with parameters $\theta$. |
| $q_\lambda(\Omega^{(1:S,1:T)}|\mathcal{G}^{(1:S,1:T)})$ | Approximate posterior distribution, i.e., inference network with parameters $\lambda$. |
| $\sigma_j$ | Temporal smoothness hyperparameter for $j \in \{\phi, \psi\}$. |
| $\mathrm{MLP}_{\theta_*}(\cdot)$ | Multilayered perception with parameters $\theta_*$. |
| $\mathrm{GRU}_{\theta_*}(\cdot)$ | Gated recurrent unit with parameters, $\theta_*$ |
| $L_{\theta_*}$ | Number of layers in multilayered perception/ gated recurrent unit |
| $\mathrm{Normal}(\cdot, \cdot)$ | Normal distribution with mean $\boldsymbol{\mu}^{(s,t)} \in \mathbb{R}^a$ and standard deviation $\boldsymbol{\mu}^{(s,t)} \in \mathbb{R}_{\geq 0}^a$ |
| $\mathrm{Categorical}(\cdot)$ | Categorical distribution with probabilities $\boldsymbol{\pi}^{(s,t)} \in \Delta^{j-1}$ for $j \in \{K, V\}$ |

Table 2: Summary of notation.

## Appendix C. Neural network implementation

In Table 3, we provide the architecture of all neural network layers used in the implementation of DBGDGM.

| Network | Neural Network | Layers | (input shape, output shape) |
|---|---|---|---|
| $\mathrm{MLP}_{\lambda_z} = \mathrm{MLP}_{\theta_z}$ | Linear Layer | 1 | $(H_\phi, K)$ |
| $\mu_{\lambda_\alpha}, \sigma_{\lambda_\alpha}$ | Embedding Layer | 1 | $(S, H_\alpha), (S, H_\alpha)$ |
| $\mathrm{GRU}_{\lambda_\phi}$ | GRU | 1 | $(2 * H_\phi, 2 * H_\phi)$ |
| $\mathrm{GRU}_{\lambda_\psi}$ | GRU | 1 | $(2 * H_\psi, 2 * H_\psi)$ |
| $\mathrm{MLP}_{\theta_\phi} = \mathrm{MLP}_{\lambda_\phi}$ | Linear Layer | 1 | $(H_\alpha, H_\phi)$ |
| $\mathrm{MLP}_{\theta_\psi} = \mathrm{MLP}_{\lambda\psi}$ | Linear Layer | 1 | $(H_\alpha, H_\psi)$ |
| $\mathrm{MLP}_{\theta_c}$ | Linear Layer | 1 | $(H_\psi, V)$ |

Table 3: Summary of neural networks and learnable embeddings used in our implementation of DBGDGM.

## Appendix D. Datasets

To create multi-subject DBG datasets, we use real fMRI scans from the UK Biobank (Sudlow et al., 2015) and Human Connectome Project (Van Essen et al., 2013). Both data sources represent well-characterized population cohorts that have undergone standardized neuroimaging and clinical assessments to ensure high quality.

**UK Biobank[2] (UKB)** The UKB dataset consists of $S = 300$ resting-rate fMRI scans (i.e. 3D image of the brain taken over consecutive timepoints) randomly sampled from the v1.3 January 2017 release ensuring an equal male/female split (i.e. sex balanced) with an age range of $44 - 57$ years. The total number of images for each scan is 490 timepoints (6 minutes duration with a repetition time of 0.74s). The dataset is minimally preprocessed following the pipeline described in Alfaro-Almagro et al. (2018).

**Human Connectome Project[3] (HCP)** The HCP dataset similarly consists of $S = 300$ sex balanced resting-state fMRI scans randomly sampled from the S1200 release with an age range of $22 - 35$ years. Only images from the first scanning-session using left-right phase encoding are used. The total number of images for each scan is $1,200$ timepoints (15 minutes duration with a repetition time of 0.72s). The dataset is minimally preprocessed following the pipeline described in Glasser et al. (2013)

**Further preprocessing** The fMRI scans from each dataset are further preprocessed to create DBGs. Firstly, each scan is transformed into a multivariate timeseries of BOLD signals using the Glasser atlas (Glasser et al., 2016) to average voxels within $V = 360$ brain regions. Next, to ensure comparability with UKB, we truncate the length of HCP timeseries to 490 timepoints. Following the commonly used sliding-window method (Calhoun et al.,

---

2. https://www.ukbiobank.ac.uk

3. https://www.humanconnectome.org

2014), we use Pearson correlation to calculate FC matrices within non-overlapping windows of length $1 < W \leq 490$ along the temporal dimension. At every window, we create an edge set of a unweighted and undirected graph with no self-edges by thresholding the top $1 \leq \epsilon < 100$ percentile values of the lower triangle of the FC matrix (excluding the principal diagonal) as connected following Kim et al. (2021). For both datasets, we choose $W = 30$ and $\epsilon = 5$ resulting in $T = \lfloor 490/30 \rfloor = 16$ graph snapshots each with $E^{(s,t)} = \lfloor (360(360 - 1)/2)(5/100) \rfloor = 3,231$ edges.

## Appendix E. Baselines

We compare DBGDGM against a range of static and dynamic unsupervised graph representation learning baseline models, all with publicly available code. In particular, we focus on baselines that are generative and can quantify uncertainty. We leave comparisons to popular deterministic baselines such as DynamicTriad (Zhou et al., 2018), DySAT (Sankar et al., 2020), and DynNode2Vec (Mahdavi et al., 2018) for future work. Furthermore, since all of the baselines were originally designed to model large single-graph datasets, we had to adapt each implementation to work with smaller multi-graph datasets.

**Variational graph auto encoder**[4] **(VGAE)** (Kipf and Welling, 2016b)  An extension of the variational autoencoder (Kingma and Welling, 2013) (VAE) for graph structured data. Specifically, VGAE uses a graph convolutional network (GCN) (Kipf and Welling, 2016a) to learn a distribution over node embeddings. Originally designed for static graphs, we train VGAE on each dynamic graph snapshot independently.

**Overlapping stochastic block model**[5] **(OSBM)** (Mehta et al., 2019)  A deep generative version of the overlapping stochastic block model (Miller et al., 2009). In particular, OSBM places a stick-breaking prior over the number of communities which allows the model to automatically infer the optimal number of communities from the data during training. Similar to VGAE, OSBM uses a GCN to parameterize the distribution over node embeddings and is designed for static graphs.

**Variational graph RNN**[6] **(VGRNN)** (Hajiramezanali et al., 2019)  An extension of VGAE for dynamic graphs. Using a modified graph RNN architecture, VGRNN is able to learn dependencies between and within changing graph topology over time. Similar to DBGDGM, the prior distribution over node embeddings is parameterized using hidden states from previous timepoints.

**Evolving latent space model**[7] **(ELSM)** (Gupta et al., 2019)  A generative model for dynamic graphs that learns node embeddings and performs community detection. In particular, node embeddings are initially sampled from a Gaussian mixture model over communities and then evolved over time using an LSTM. Unlike the previous baselines, ELSM does not use a GNNs to parameterize model distributions.

---

4. https://github.com/tkipf/gae

5. https://github.com/nikhil-dce/SBM-meet-GNN

6. https://github.com/VGraphRNN/VGRNN

7. https://github.com/sh-gupta/ELSM

**vGraph**[8] **(VGRAPH)** (Sun et al., 2019)   Similar to DBGDGM, VGRAPH simultaneously learns node embeddings and community assignments by modeling nodes as being generated from a mixture of communities. The generative process of VGRAPH also relies on edge information. Since VGRAPH only models static graphs, we train it on each dynamic graph snapshot independently.

**Common neighbors (CMN)**   In light of recent work demonstrating that heuristic methods are able to outperform deep-learning based models on dynamic link prediction tasks (Skarding et al., 2022; Poursafaei et al., 2022), we include our own heuristic-based generative model baseline. More formally, let $\boldsymbol{\pi}_{v_i}^{(t)} \in [0,1]^V$ denote a vector of Jaccard index scores for node $v_i^{(t)} \in \mathcal{V}$ with all other nodes $v_j^{(t)} \in \mathcal{V}$ for $i \neq j$. The Jaccard index between two nodes $v_i^{(t)}$, $v_j^{(t)} \in \mathcal{V}$ is defined $|\Gamma(v_i^{(t)}) \cap \Gamma(v_j^{(t)})|/|\Gamma(v_i^{(t)}) \cup \Gamma(v_j^{(t)})|$ where $\Gamma(v_i^{(t)})$ denotes the set of neighbors of node $v_i^{(t)}$. We define the probability of node $v_i^{(t)}$ having a linked neighbor $v_j^{(t)}$ at snapshot $t$ as

$$p(v_j^{(t)}|v_i^{(t)}) = \text{Categorical}(\boldsymbol{\pi}_{v_i}^{(t-1)}). \tag{15}$$

This simple generative model captures the intuition that nodes are more likely to form links if they had common neighbors in a previous snapshot.

## Appendix F. Implementation details

**Software and hardware**   All models are developed in Python 3.7 (Python Core Team, 2019) using scikit-learn 1.1.1 (Pedregosa et al., 2011), PyTorch(Paszke et al., 2019), and numpy 1.1.1 (Harris et al., 2020). Statistical significance tests are carried out using deep-significance 1.1.1 (Ulmer et al., 2022). Experiments are performed on a Linux server (Debian 5.10.113-1) with a NVIDIA RTX A6000 GPU with 48 GB memory and 16 CPUs.

**Training and testing**   All baselines are implemented as per the original paper and/or code repository given in Appendix E. For the static graph baselines VGAE, OSBM, VGRAPH we train on each snapshot independently and use the node and/or community embeddings at the last training snapshot to make predictions.

**Hyperparameter optimization**   We use model and training hyperparameter values described in the original implementation of each baseline as a starting point for tuning on the validation dataset. Since searching for optional values for each hyperparameter configuration is outside the scope of the paper, we focus mainly on tuning the dimensions of hidden layers. For DBGDGM, we use a learning rate of 1$e$-4 with a weight decay of 0. We choose the number of communities $K \in \{3, 6, 8, 12, 16, 24\}$ based on lowest average validation NLL (see Figure 4). In the generative model, we fix the temporal smoothness hyperparameters $\sigma_\phi = \sigma_\psi = 0.01$. In the inference network, we fix the number of layers for all NNs to $L_\phi = L_\psi = L_z = 1$. For the Gumbel-softmax reparameterization trick we anneal the softmax temperature parameter starting from a maximum of 1 to a minimum of 0.05 at a rate of 3$e$-4. Finally, we train all models for $1,000$ epochs using early-stopping with a patience of 15 based on the lowest validation NLL.

---

8. https://github.com/fanyun-sun/vGraph

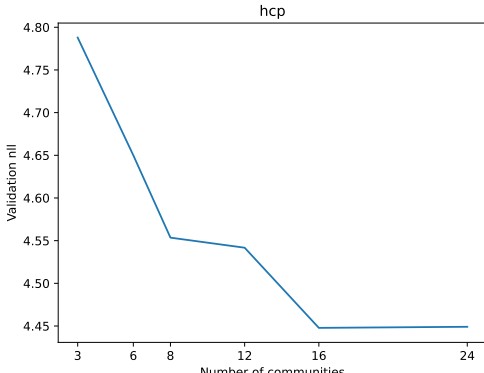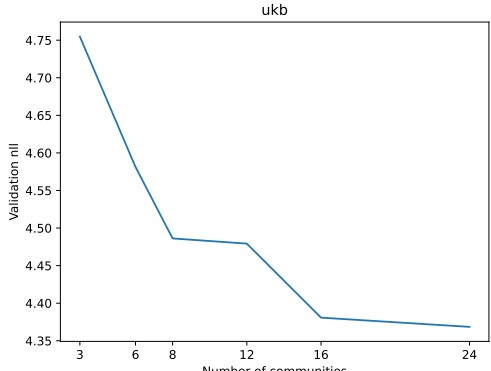

Figure 4: Elbow plot for finding the optimal number of communities $K$.

## Appendix G. Task description

**Graph reconstruction** The aim of the graph reconstruction task is to assess each model's ability at learning the underlying dynamic brain graph data generating distribution. A model that has good graph reconstruction ability is able to recreate the complete dynamic brain graph structure and it's (statistical) properties as accurately as possible.

For evaluations metrics we use negative-log likelihood (NLL) and mean squared error (MSE) of the temporal node degree between the ground truth and reconstructed graphs. More specifically, NLL measures how likely it is that a model has generated the test data, it is commonly used to evaluate deep generative models (Grover et al., 2019; You et al., 2018). On the other hand, MSE computes the distance between the temporal node degree (Nicosia et al., 2013; Bassem, 2019) of ground truth and reconstructed graphs from the test dataset. We add a definition of the latter metric to the appendices.

**Dynamic link prediction** The aim of dynamic link prediction is to predict edges between nodes at one or more future snapshots using graph structure data up to and including the current snapshot. We use either the node or node and community embeddings from the last training snap shot to make predictions on all future test snap shots, following (Grover et al., 2019; You et al., 2018; Nicosia et al., 2013; Bassem, 2019). For each baseline, we follow the exact link prediction implementation as described in the original papers.

For evaluation metrics use we area under the receiver operative curve (AUROC) and average precision (AP), both of which are commonly used evaluation metrics for dynamic link prediction (Gracious et al., 2021). More specifically, AUROC computes the area under the true-positive rate vs false-positive rate curve for various values of classification threshold. AP, on the other hand, summarizes the precision-recall curve as the weighted mean of precisions for various values of classification threshold, with the increase in recall from the previous threshold used as the weight. For each graph at every test snap shot, we assume all known edges are true and sample the same number of non-edges as false, as is common practice (Gracious et al., 2021; Pareja et al., 2020; Hajiramezanali et al., 2019; Poursafaei et al., 2022). We compute the per snapshot AUROC and AP and report the average. We do not consider other performance metrics such as accuracy, precision, recall, or F1-score

since they require a proper confidence threshold to be specified which can lead to unfair comparison across different models (Yang et al., 2015).

**Graph classification**   The aim of graph classification is to predict a label at the graph-level. In our paper, this label is set to biological sex. For evaluation, since the DBG dataset is 50-50 male-female balanced, we use accuracy with a classification threshold of 0.5.

## Appendix H. Interpretability analysis

Evidence from fMRI studies suggests complex community structure of brain graphs (Ting et al., 2020; Martinet et al., 2020a). These communities often correspond to groups of anatomically neighboring and/or functionally related brain regions that are engaged in specialized information processing. In order to interpret the community structure learnt by DBGDGM, for each community we create a community-node score vector $\bar{\psi}_k \in [0, 1]^N$ by averaging sampled community embeddings over subjects and timepoints following

$$\bar{\psi}_k = \frac{1}{ST} \sum_{s=1}^{S} \sum_{t=1}^{T} \text{Softmax}\big(\text{MLP}_{\lambda_c}(\psi_k^{(s,t)})\big) \tag{16}$$

and keep the top 10% highest scoring nodes. We use these thresholded nodes to calculate the proportion of overlap between each community and known functional connectivity networks (FCNs) from Ji et al. (2019). More specifically, the colored sections for each community in Figure 3 represent the proportion of nodes in each community which belong to a FCN.

| Abbreviation | Functional network |
| --- | --- |
| AUD | Auditory network |
| CON | Cingulo-opercular network |
| DAN | Dorsal-attention network |
| DMN | Default mode network |
| FPN | Frontoparietal network |
| LAN | Language network |
| ORA | Orbito-affective network |
| PMM | Posterior-multimodal network |
| SMN | Somatomotor network |
| VIS1 | Visual network 1 |
| VIS2 | Visual network 2 |
| VMM | Ventral-multimodal network |

Table 4: Functional connectivity networks (FCNs) from Ji et al. (2019).

## Appendix I. Training time and parameter count

In Table 5, we provide the parameter counts, multiply–accumulate (MAC) operations and time per epoch for all models evaluated on the HCP dataset. We use the DeepSpeed

library[9] to obtain the parameter count, MACs, and time taken to train for one epoch. We observe that DBGDGM is the fastest dynamic method to train and second fastest across all baselines. In addition, we observe that our method has the lowest MACs in a forward pass. On the other hand, DBGDGM has the highest number of parameters.

| Model | Dynamic | Complexity | Computation | Time |
|---|---|---|---|---|
| VGAE | ✗ | 50.34 k | 10.79 GMACs | 468.47 s |
| VGRNN | ✓ | 253.89 k | 53.69 GMACs | 586.43 s |
| ELSM | ✓ | 905.25 k | 702.54 MMACs | 508.89 s |
| OSBM | ✗ | 52.40 k | 10.09 GMACs | 369.91 s |
| VGRAPH | ✗ | 94.58 k | 47.30 GMACs | 906.88 s |
| DBGDGM | ✓ | 6.70 m | 6.20 GMACs | 425.00 s |

Table 5: Parameter count, MACs in a forward pass and time per epoch for all models.

---

9. https://github.com/microsoft/DeepSpeed

