# OpenReview forum: "DBGDGM: Dynamic Brain Graph Deep Generative Model"
_MIDL.io/2023/Conference — MIDL 2023 Oral_

### Official Review · Reviewer_N5RA · 2023-02-04

**Confidence:** 4
**Preliminary Rating:** 4

**Summary:**

The paper proposes a deep generative model for clustering brain regions into dynamically evolving communities in rs-fMRI data. Specifically, regions of the brain are represented as nodes in a graph whose embeddings evolve over time and are distributed across a small set of communities common to all subjects. The distribution and community memberships are parameterized using deep neural networks whose parameters can be estimated in a data-driven fashion.

Experimental validation is performed on the HCP dataset against a variety of dynamic connectivity baselines to evaluate tasks of graph generation, dynamic link prediction, and graph-based classification. A qualitative comparison of the learned communities is also performed to validate their consistency with neuroscientific findings in the literature

**Strengths:**

The neural network parameterization for the community detection problem and variational formulation for training and inference is an interesting and possibly novel methodological contribution to the field.  The main motivation is clearly articulated in the context of the shortcomings of existing frameworks.  Extensive comparisons have been performed against several dynamic and static connectivity modeling baselines on several graph prediction tasks, many of which the model performs well on.


**Weaknesses:**

1. The description of the tasks used for evaluation is very hard to parse from the description.

(a) For example, what is the difference in the setup of the graph reconstruction and dynamic link prediction task conceptually? Are positive and negative values in the FC correlation features (ground truth connectivity) treated the same for evaluation?

(b)  The description of the graph classification task mentions "To predict graph labels, we average node embeddings per subject for the baselines and the community embeddings for DBGDGM before training a SVM using 10-fold cross-validation. Does this mean that unsupervised representation learning is first performed on the entire dataset or is the learning a part of the cross validation loop? Would the averaging of the node/community embeddings (before training the classifier) be prone to issues of over-smoothing, as in the case of graph convolutional networks?

2. "We split both datasets into 80/10/10% training/validation/test data along the time dimension." I am not sure I understand this statement entirely. Does this mean that the folds are *not* stratified across subjects? Is the cross validation set up differently for the graph labeling setup? Per random seed, does the split change or is the only change in the initialization of the models? Are the communities uncovered consistent across subsamples of the data?

**Deanonymize Review:**

no

**Detailed Comments:**

1. The methods section introduces a lot of notation/indices, making it cumbersome to reference across pages. A small suggestion would be to include a table of notation in the appendix to improve readability.

2. It would be great if the authors could comment on the relative training time for this framework in comparison with the other unsupervised learning frameworks.

**Paper Type:**

methodological development

**Questions To Address In The Rebuttal:**

 Please focus on addressing comments in the weaknesses section by providing a more detailed description of the setup for the three tasks, perhaps as a separate section from the one describing the evaluation metrics.

---

### Official Review · Reviewer_GDEN · 2023-02-05

**Confidence:** 2
**Preliminary Rating:** 4

**Summary:**

The authors propose the use of a deep generative model to learn from dynamic brain graphs. The generative model treats node embeddings as samples from a random process over community clusters that are able to change over time. The authors use the proposed model to learn from fMRI data to perform of graph reconstruction, dynamic link prediction and graph classification.

**Strengths:**

The paper is generally well written and easy to understand, the experimental results presented show that the proposed model is superior to many other baselines for the task of graph reconstruction and link prediction. Additionally, code is to be made publicly available.

**Weaknesses:**

The paper could benefit from a more introduction in the fMRI data to better motivate the use of dynamic brain graphs over static brain graphs. It is not immediately clear to me that incorporating dynamic information improves graph classification (for the proposed model, VGRNN and ELSM)

**Deanonymize Review:**

no

**Paper Type:**

methodological development

**Questions To Address In The Rebuttal:**

From my understanding, from equations two and three, is it the case that we are sampling node and community embeddings, each node embedding is then mapped to a community via equation four. And equation five describes the process of sampling links (target nodes for a source node) based on the community assignment?

Please kindly add more detail to the caption of Figure 3 so that it can be more easily understood in isolation.

---

### Official Review · Reviewer_cQpS · 2023-02-07

**Confidence:** 4
**Preliminary Rating:** 4
**Recommendation:** Poster

**Summary:**

This paper investigates a new method of analyzing resting-state fMRI images in experiments of graph generation and graph classification tasks. Compared to existing methods, it focuses on extracting the temporally dynamic information of brain connectivity in both node and community levels. The idea could be a valuable addition to existing work of deep-learning-based spatio-temporal analysis on fMRI data.

**Strengths:**

This paper proposes a novel graph generative model that shows outstanding results in graph generative tasks. The method of jointly learning static graph embedding and dynamic node/community embeddings is innovative. Using two publicly available datasets improves the reliability of work. The explanation of general algorithm is sufficient.


**Weaknesses:**

1. As already stated in the paper, the performance of the proposed model is similar to several baseline approaches in the graph classification task. Some discussions of the model performance gap between generation and classification task would be good.

2. The presentation of materials lacks graphical content for ease of understanding, such as graphs for overall pipeline or network implementation details.


**Deanonymize Review:**

no

**Detailed Comments:**

Typo
Page 6 line 1: ‘import’ should be ‘input’


**Paper Type:**

methodological development

**Questions To Address In The Rebuttal:**

1. What is your justification for learning a time-invariant graph embedding? From your problem setting, it seems natural to learn a dynamic graph embedding for each temporal ‘graph snapshot’ same as node and community embeddings.

2. In your explanation of graph generation, the initial values of node and community embedding at t=0 are set to equal to graph embedding. Does this enforce graph, node, and community embeddings to have the same dimensionality?

3. For the experiments of baseline models, have you tried aggregating sequential output instead of using just the last snapshot output?

4. Would you mind providing some comparisons of model parameter size and training time with regards to baseline approaches? It would be useful for understanding your model performance comprehensively.

---

### Meta-Review · Area_Chair_E9u3 · 2023-02-25

**Recommendation:** Accept (Poster)
**Confidence:** 4

**Metareview:**

The authors propose a dynamic brain graph deep generative model that treats node embeddings as samples from a random process over community clusters that are able to change over time. The authors use the proposed model to learn from fMRI data to perform of graph reconstruction, dynamic link prediction and graph classification.

Reviewers agree on the interest and novelty of the proposed approach, potential reproducibility of the work, and strength of multiple comparisons in the experiments. Reviewer concerns regarding clarity of certain experimental details were addressed by authors. Reviewers also noted the similar performance of the proposed method compared to many existing approaches, to which authors stress the interpretability of their approach in terms of functional network/community impact.

Given the interesting approach of incorporating a model for functional communities into the dynamic graph generation process and potential demonstrated by the experiments, I agree with the reviewers' ratings and recommend acceptance of this work.

---

### Meta-Review · Program_Chairs · 2023-03-01

**Recommendation:** Accept (Oral)
**Confidence:** 5

**Metareview:**

The PC discussion concluded that the paper has enough technical novelty, combined with clinical application, that the paper could be moved to an Oral presentation.